# Clinical use and indications for head computed tomography in children presenting with acute medical illness in a low- and middle-income setting

Pamela Rudo Machingaidze[1,2], Heloise Buys[1,2]*, Tracy Kilborn[2,3], Rudzani Muloiwa[1,4]

**1** Department of Paediatrics and Child Health, University of Cape Town, Cape Town, South Africa, **2** Red Cross War Memorial Children's Hospital, Cape Town, South Africa, **3** Department of Radiology, University of Cape Town, Cape Town, South Africa, **4** Department of Paediatrics, Groote Schuur Hospital, Cape Town, South Africa

* heloise.buys@uct.ac.za

**Data Availability Statement:** All relevant data are within the manuscript and its Supporting Information files.

## Abstract

### Background

Computed tomography (CT) imaging is an indispensable tool in the management of acute paediatric neurological illness providing rapid answers that facilitate timely decisions and interventions that may be lifesaving. While clear guidelines exist for use of CT in trauma to maximise individual benefits against the risk of radiation exposure and the cost to the health-care system, the same is not the case for medical emergency.

### Aims

The study primarily aimed to retrospectively describe indications for non-trauma head CT and the findings at a tertiary paediatric hospital.

### Methods

Records of children presenting with acute illness to the medical emergency unit of Red Cross War Children's Hospital, Cape Town, over one year (2013) were retrospectively reviewed. Participants were included if they underwent head CT scan within 24 hours of presentation with a non-trauma event. Clinical data and reports of CT findings were extracted.

### Results

Inclusion criteria were met by 311 patients; 188 (60.5%) were boys. The median age was 39.2 (IQR 12.6–84.0) months. Most common indications for head CT were seizures (n = 169; 54.3%), reduced level of consciousness (n = 140;45.0%), headache (n = 74;23.8%) and suspected ventriculoperitoneal shunt (VPS) malfunction (n = 61;19.7%). In 217 (69.8%) patients CT showed no abnormal findings. In the 94 (30.2%) with abnormal CT results the predominant findings were hydrocephalus (n = 54;57.4%) and cerebral oedema (n = 29;30.9%). Papilloedema was more common in patients with abnormal CT (3/56; 5.4%)

**Funding:** The author(s) received no specific funding for this work.

**Competing interests:** The authors have declared that no competing interests exist.

compared with none in those with normal CT; P = 0.015; while long tract signs were found in 42/169 (24.9%) and 23/56 (41.1%) of patients with normal and abnormal CT findings, respectively; P = 0.020. Post-CT surgery was required by 47(15.1%) of which 40 (85.1%) needed a ventricular drainage. A larger proportion of patients with VPS (25/62; 40.3%) required surgery compared to patients without VPS (22/249; 8.8%; P<0.001).

## Conclusion

A majority of head CT scans in children with medical emergency with acute neurological illness were normal. Patients with VPS constituted the majority of patients with abnormal CT scans that required subsequent neurosurgical intervention. Evidence-based guidelines are required to guide the best use of head CT in the management of children without head trauma.

## Introduction

Computed tomography (CT) is an indispensable tool in the management of paediatric illness, particularly in the acute diagnosis of medical or surgical intracranial pathology. It can give answers quickly, allowing potentially lifesaving decisions to be made urgently [1–3]. A number of studies show that CT head or brain is the most common CT examination in children [4–6]. This contrasts with older age groups in which abdominal and pelvic CT scans predominate [7, 8].

The benefits of CT must be weighed against the risks to the patient and health care system. CT carries potential risk of malignancy because of its associated ionizing radiation. The Food and Drug Administration (FDA) has stated that the probability of getting cancer from the doses used in head CT in children is thought to be very small; however, no amount of radiation can be considered absolutely safe [9]. CT imaging also carries infrastructural costs including radiographer and radiologist time and potential use of contrast media. In younger patients, sedation may be needed to achieve optimal imaging results, and this may require the support of the anaesthetic department.

While there are clear guidelines for CT following head trauma, the same is not the case for CT head in non-trauma medical emergency [9, 10]. This means that requests for CT scan are generally made on unclear bases which has significant implications when there are significant resource limitations such as in low and middle income country (LMIC) settings.

There are some published data on the use of CT in Africa in the management of meningitis and paediatric seizures [11]. However, data on guiding the use of CT in acute paediatric medical illness in resource-limited settings are lacking [12]. Our study aimed to describe the clinical use of emergency head CT scans at a tertiary paediatric hospital in a low- and middle-income country setting. The primary outcomes of interest were indications for and findings of head CT imaging in children presenting for acute medical care, as well as to establish baseline characteristics and interventions performed post CT scanning. Secondarily, we compare frequencies of presenting factors between participants with normal and abnormal findings on CT.

## Methods

A retrospective observational study was done on a cohort of children presenting with acute medical illness requiring CT scan of the brain. A list of CT scans performed over one year was

compiled from the radiology department's Picture Archiving and Communication System (PACS) of the Red Cross War Memorial Children's Hospital (RCWMCH), Cape Town, South Africa. RCWMCH is a tertiary referral hospital servicing a paediatric population of about 1.5 million children. At this hospital there is a separate trauma unit which is not linked to the medical emergency unit (MEU). Indications for computed tomography (CT) and subsequent management in the trauma unit follow well established international trauma guidelines. Although the hospital has clinical guidelines for performing CT in the MEU, these are relatively flexible and there are frequent deviations from these guidelines.

All children seen in the MEU from 1 January 2013 to 31 December 2013 who underwent brain CT imaging within 24 hours of consultation or admission were eligible for inclusion. Subjects were excluded if referral for CT was not done in the MEU as part of their assessment; injured children are seen in a separate trauma unit at this institution. Demographic data were extracted from records, and indications for CT as well as clinical presentation were documented for each child. CT findings as independently reported by a junior radiologist and reviewed by an experienced paediatric radiologist were noted. The level of experience and seniority of the clinician requesting the scan was also assessed.

Head CT scan findings were classified as normal (clinically insignificant) if a first-time scan was reported as normal or where no interval change on CT findings of a participant with known pre-existing abnormality on CT was found. CT findings reported as abnormal in first-time CTs or where interval change had occurred in subjects with known abnormal findings on previous CT were regarded as abnormal (clinically significant).

Data were analysed using STATA software version 13 (STATA Corporation, College Station, Texas, USA). Categorical variables were represented as proportions using percentages. Continuous variables were summarised using medians with interquartile ranges (IQR). Categorical variables were compared using Pearson's Chi-square or Fisher's exact tests as appropriate. We estimated that over a one-year period about 250 to 350 children seen in the medical emergency unit would make inclusion criteria. Based on this sample size, for outcomes ranging from 5% to 20%, the precision will fall within 3% of the point estimate.

Approval for the study was granted by the Research Ethics Committee of the University of Cape Town, and the administration of RCWMCH; Ethics reference HREC/Ref: 087/2015.

## Results

### Baseline characteristics of included study subjects

A total of 311 subjects, representing 9.4% of the 3300 CT scans done in the hospital in 2013 met inclusion criteria (Fig 1). The cohort included 188 boys (60.5%). The median age of the group was 39.2 (IQR 12.6–84.0) months and ranged from two and a half weeks to 15 years-of-age. There were 62 (19.9%) patients who had cerebrospinal fluid (CSF) shunts, one of whom had a ventriculopleural shunt, three had cystoperitoneal shunts and the rest had indwelling ventriculoperitoneal shunts. In addition to having CSF shunts, 25 (40.3%) of shunted patients also had a diagnosis of epilepsy.

For 225 (72.3%) of the study subjects this was their first head CT scan. In 74 (86.0%) out of 86 patients for whom the 2013 scan was a repeat, the number of previous scans could be determined. The median number of previous scans was four (IQR 2–7), ranging from one to 22 scans for a total of 365 previous scans. Individuals with CSF shunts accounted for 62 (72.1%) of patients with previous head CTs and 322 (88.2%) of the total known number of previous scans. The total number of previous scans could not be ascertained for 12 of the patients (Table 1).

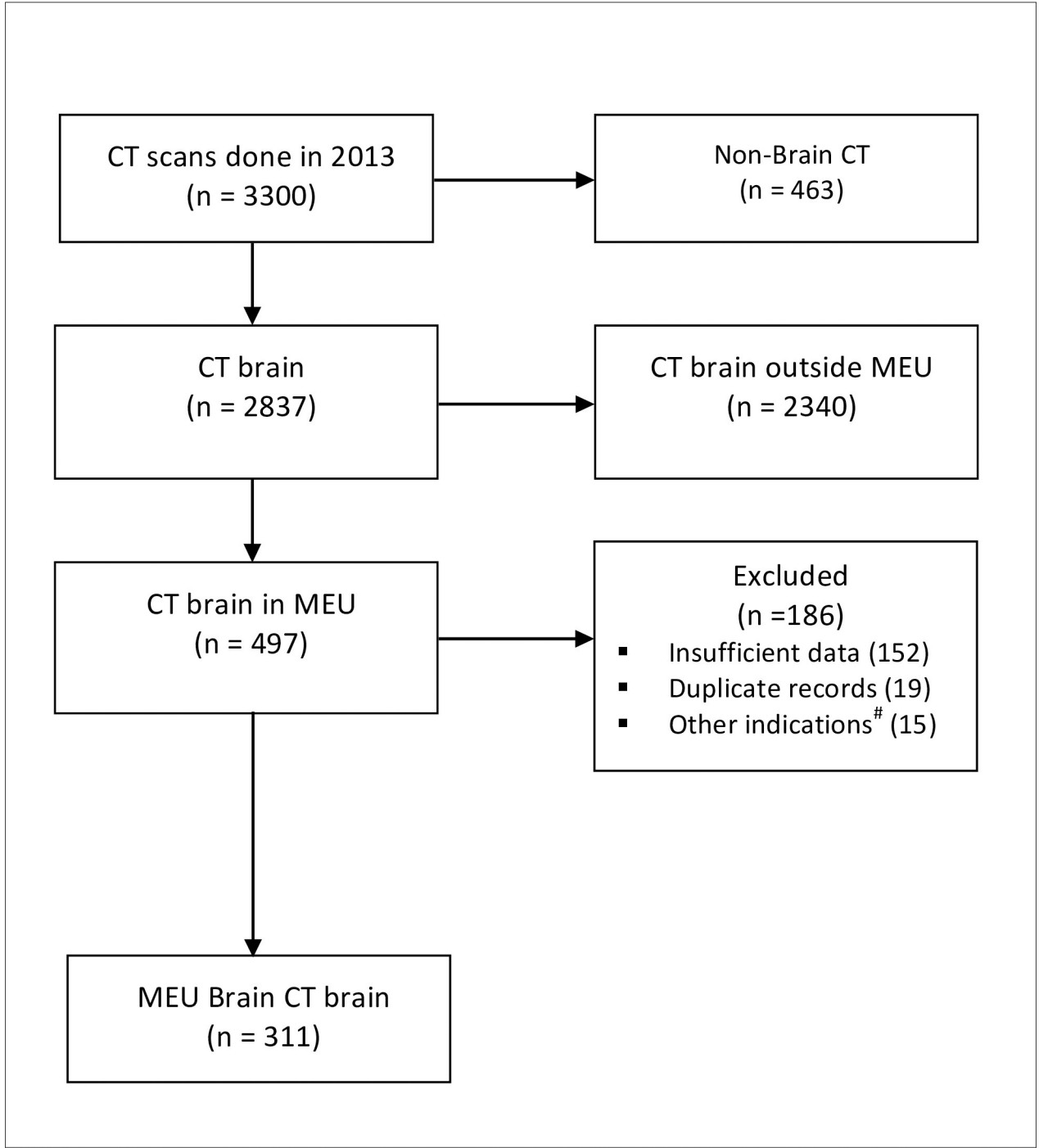

**Fig 1. Flow diagram of sample selection for enrolment.** #CT done for ophthalmology and otorhinolaryngology purposes. CT–computed tomography. MEU–medical emergency unit.

### Indications for head CT

Ninety-six (30.9%) decisions to perform CT were made by senior staff, that is, paediatric consultants and senior registrars. Most requests (n = 163;52.4%) were made by junior staff, comprising junior paediatric and neurosurgical registrars, medical officers and interns. For the

**Table 1. Baseline characteristics of study participants.**

| Baseline characteristic | N = 311 |
|---|---|
| **Age** | |
| Median IQR months | 39.2 (12.6–84.0) |
| | **n (%)** |
| **Sex** | |
| Female | 123 (39.5) |
| Male | 188 (60.5) |
| **Origin** | |
| Referred | 191 (61.4) |
| Self-referred | 101 (32.5) |
| Unknown | 19 (6.1) |
| First scan | 225 (72.3) |
| Repeat scan | 86 (27.7) |
| CSF shunt in situ | 62 (19.9) |
| No CSF shunt in situ | 249 (80.1) |
| Known epilepsy diagnosis | 54 (17.4) |
| Previous seizures | 30 (9.7) |
| **HIV status** | |
| Unexposed uninfected | 158 (50.8) |
| Exposed uninfected | 31 (10.0) |
| Infected | 7 (2.2) |
| Unknown HIV status | 115 (37.0) |

IQR–interquartile range, CSF–cerebrospinal fluid.

remaining 52 (16.7%) patients it could not be established from the patient record who had ordered the scan.

The median time from ordering the head CT to performing it was 63 (IQR 38–112) minutes, ranging from 10 minutes to 21.7 hours.

Indications for CT have been shown in **Table 2** in descending order of frequency. The majority of study subjects had more than one indication.

**Table 2. Indications for head CT in children presenting with acute medical illness.**

| | All (n = 311) | Abnormal CT (n = 92) |
|---|---|---|
| Indication# | n (%) | n (%) |
| Seizures | 169 (54.3) | 43 (46.7) |
| Impaired level of consciousness | 140 (45.0) | 42 (45.7) |
| Headache | 74 (23.8) | 28 (30.4) |
| Suspected VPS pathology | 61 (19.7) | 28 (30.4) |
| Focal neurological signs | 42 (13.5) | 14 (15.2) |
| Suspected raised intracranial pressure | 26 (8.4) | 13 (14.1) |
| Suspected hydrocephalus | 18 (5.8) | 7 (7.6) |
| Suspected tuberculous meningitis | 8 (2.6) | 2 (2.2) |

CT = computed tomography, #majority of study subjects had more than one indication. VPS–ventriculoperitoneal shunt.

One child, a two-year-old female with hydrocephalus secondary to neonatal meningitis and a ventriculoperitoneal shunt (VPS) in situ, had the highest number of head CT scans. She was scanned 5 times during 2013 and 22 times in her lifetime. The same patient underwent three VPS revisions in 2013.

The most common indication for CT was seizures (n = 169; 54.3%) with 63 (37.5%) of the patients having generalised seizures, 59 (35.1%) focal and eight (4.8%) classified as atypical. In 28 (16.7%) both focal and generalised seizures co-existed while the type of seizure was unknown in 10 (6.0%). The median seizure duration was 15 (IQR 5–30) minutes, with the longest seizure lasting 4 hours (240 minutes) and the shortest less than a minute. In 49 (37.7%) of the participants, a diagnosis of status epilepticus (SE) was made by the attending clinician, defined in this paper as a seizure lasting more than 30 minutes. The presenting seizure was the first seizure episode for 53 (31.4%) of the patients while 54 (17.4%) had a pre-existing diagnosis of epilepsy. Of the patients who had prior head CT scans, 28 were known with a diagnosis of epilepsy. Twenty-eight (16.6%) patients were documented as having febrile seizures.

## Findings on CT scan

A total of 219 (70.4) patients out of 311 had normal CT findings. These were composed of 169 with normal first-time scans and 50 repeats with no interval change. The 92 (29.6%) with abnormal CT findings were composed of 56 abnormal fast time scans and 36 with interval change.

Hydrocephalus was the most common abnormal finding with 54 (58.7%) of the 92 abnormal CTs showing this finding. Twenty-nine patients with CSF shunts presented with hydrocephalus on CT scan. This was followed by cerebral oedema in 29 (31.5%). The other abnormal CT findings are shown in **Table 3**. There were 29 (46.8%) of the 62 patients with VPS that had at least one abnormal finding on CT (interval change) compared to 63 (25.3%) of the 249 without VPS (i.e. no new findings or interval change if previously scanned); P = 0.001.

Fifteen (27.8%) of the 54 patients with a known diagnosis of epilepsy had normal findings on CT while 21 (38.9%) had known pre-existing pathology which was unchanged. A total of 18 (33.3%) patients with epilepsy had abnormal CT findings of which six (33.3%) were findings on first time CT and 12 (66.7%) interval change on pre-existing CT pathology.

**Table 3. Computed tomography head findings in children with acute medical illness.**

| Finding on CT# | All n = 311 n (%) | VPS patients n = 62 n (%) | No VPS n = 249 | P value |
|---|---|---|---|---|
| Normal | 219 (70.4) | 33 (53.2) | 186 (74.7) | **0. 001** |
| Hydrocephalus | 54 (17.4) | 29 (46.8) | 25 (10.0) | **< 0.001** |
| Cerebral oedema | 29 (9.3) | 2 (3.2) | 27 (10.9) | 0.085 |
| Space occupying lesion | 19 (6.1) | 0 (0) | 19 (7.6) | **0.18** |
| Cerebral atrophy | 16 (5.1) | 7 (11.3) | 9 (3.7) | **0.025** |
| Meningitis* | 12 (3.9) | 0 (0) | 12 (4.9) | 0.134 |
| Infarct | 11 (3.5) | 2 (3.2) | 9 (3.6) | 1.000 |
| Surface collection | 10 (3.2) | 5 (8.1) | 5 (2.0) | **0.030** |
| Haemorrhage | 4 (1.3) | 1 (1.6) | 3 (1.2 | 1.000 |
| Thrombosis | 3 (1.0) | 0 (0) | 3 (1.2) | 1.000 |

CT = computed tomography, VPS = ventricular peritoneal shunt,

# some study subjects had more than one abnormal finding.

*meningitis–basal or, leptomeningeal enhancement, or subdural hygroma, Bold typeface = P<0.05.

Frequency of abnormal or clinically significant findings was slightly higher in patients who presented without seizures compared to those with seizures with 49 (34.5%) out of 142 and 43 (25.4%) out of 169 respectively; P = 0.081. In patients with seizures lasting more than 15 minutes, 38/86 (44.2%) scanned for the first time, had normal findings versus 10/24 (41.7%) with abnormal findings (P = 0.826); of the patients receiving a repeat scan 10/15 (66.7%) had no interval change while 2/5 (40.0%) had new findings (P = 0.292).

Long tract signs were more frequent in patients with an abnormal first CT scan with 23 out of 56 (41.1%) compared with 42/169 (24.9%) in those with normal CT; P = 0.020. Papilloedema was absent in patients with a normal CT at first scan (**Table 4**).

Abnormal CT findings on current CT were found in 31 (50.0%) out of 62 patients with CSF shunts compared to 61 (24.5%) out of 249 in those without shunt; P<0.001.

Of the 169 patients with seizures, 151 (89.3%) did not have CSF shunts. In that cohort, 44/151 (29.1%) had abnormal findings on CT scan.

## Management and outcome

A total of 160 LPs were performed on 158 patients (2 patients had LP before and after scan). Of this group, 21 (13.1%) LPs were performed before CT scan and 139 (86.9%) after CT scan. In 70 (50.4%) patients with suspected meningitis who had LP post CT scan, CT was done first to exclude space-occupying lesions, non-communicating hydrocephalus or raised intracranial pressure, contraindications for LP. Only two (2.9%) of these had CT findings that precluded LP.

Forty-seven patients (15.1%) of the 311 had interventions based on CT scan findings. **Table 5**

Intervention was indicated in 25 (40.3%) of the 62 patients with CSF shunts compared to 22 (8.8%) of the 249 without CSF shunts; p<0.001. VPS revisions were carried out on two patients diagnosed with shunt sepsis and blocked shunt respectively although CT findings revealed no interval changes.

Eight (2.6%) patients, of whom two had normal CT, died during the admission. Causes of death were as follows: intracerebral haemorrhage due to undetermined causes, severe pneumococcal meningitis, suspected pineal mass, meningitis with subsequent cerebral herniation, complicated tuberculous meningitis and VPS malfunction with hydrocephalus. For the two with normal CT death followed severe pneumonia in one, and acute liver failure secondary to acute hepatitis A infection in the other.

**Table 4. Comparison of CT findings in patients with first and repeat CT by clinical presentation.**

| Clinical Presentation | First Computed Tomography n (%) | | | Repeat Computed Tomography n (%) | | |
|---|---|---|---|---|---|---|
| | Normal n = 169 | Abnormal n = 56 | P | No change n = 50 | New findings n = 36 | P |
| Impaired LOC | 84 (49.7) | 35 (62.5) | 0.096 | 27 (54.0) | 16 (44.4) | 0.382 |
| Nausea or vomiting | 41 (24.3) | 21 (37.5) | 0.055 | 22 (44.0) | 22 (61.1) | 0.117 |
| Papilloedema | 0 (0.0) | 3 (5.4) | **0.015** | 1 (2.0) | 3 (8.3) | 0.304 |
| Generalised seizure | 61 (36.1) | 16 (28.6) | 0.304 | 9 (18.0) | 5 (13.9) | 0.610 |
| Headache | 28 (16.6) | 11 (19.6) | 0.598 | 18 (36.0) | 17 (47.2) | 0.296 |
| Long tract signs[#] | 42 (24.9) | 23 (41.1) | **0.020** | 15 (30.0) | 11 (30.6) | 0.956 |
| Focal seizure | 53 (31.4) | 20 (35.7) | 0.546 | 10 (20.0) | 4 (11.1) | 0.271 |
| Focal neurology | 18 (10.7) | 8 (14.3) | 0.461 | 10 (20.0) | 6 (16.7) | 0.695 |

LOC = level of consciousness; CT = computed tomography; Bold typeface = P<0.05;

# Increased tone and brisk reflexes.

Table 5. **Interventions based on CT scan findings.** n = 47.

| Surgical intervention | N (%) |
|---|---|
| Cerebrospinal fluid shunt | 40 (85.1%) |
| Surgical drainage of abscess or fluid collection | 4 (8.5%) |
| Therapeutic lumbar puncture/ventricular tap | 3 (6.4%) |

CT = computed tomography.

## Discussion

Our study shows that the majority of children who present with acute medical illness and undergo emergency head CT have no clinically significant findings on CT. Unsurprisingly, children with VPS were significantly likely to have abnormal findings and to require surgical intervention following CT.

Hydrocephalus (HCP) was the most common abnormal CT finding, most likely reflecting the number of patients (20%) with CSF shunts who made up a large proportion of those undergoing the investigation and requiring intervention after imaging. These patients were also more likely to be scanned repeatedly. Patients with CSF shunts have previously been reported to have more investigations and surgical procedures in their lifetime [13]. In our study, children with shunts were significantly likely to required surgical intervention, compared to children without (40% versus 9%). This finding was in keeping with other studies in which patients with VPS presenting to the emergency department required surgical intervention [14]. This is not surprising as shunts are prone to numerous complications such as mechanical obstruction, malfunction, fracture, infection, migration and excessive CSF drainage [15].

Children with abnormal findings on first CT were more likely to present with abnormal clinical findings although a statistically significant association was manifest only with the presence of papilloedema or long tract signs. There was also a moderately strong association with nausea and vomiting. A cohort study performed on an adult American population found that in addition to altered mental status and focal neurology, papilloedema was a significant predictor of new intracranial pathology on CT scan [16].

Seizures were the most common indication for head CT in this study. In 16.6% of the patients presenting with seizures a diagnosis of complex febrile seizures was made; all their CT imaging was normal. Other studies have noted that patients with complex febrile seizures were more likely to receive an extensive workup, including a CT scan; however, none of the head CT scans performed showed significant findings that necessitated intervention or guided therapy [17]. In our study, out of the 151 patients with seizure and did not have CSF shunts, less than a third (29%) had abnormal findings on CT. The risk of abnormal CT findings was however not associated with the duration of seizures or whether the patients presented with focal or generalised seizures. Swingler et al., also concluded that routine CT imaging in children with recent onset partial seizures did not meaningfully change clinical management [11], concurring with a year-long retrospective review of children presenting with first-onset seizures to the ED who underwent brain CT, excluding patients with simple febrile seizures [18]. Of the 66 patients, 14 (21.2%) had abnormal results. Another study by Allen and Jones, assessed children with epilepsy presenting with breakthrough seizures and undergoing head CT scanning [18]. None of the scans had acute findings and they were all discharged from the emergency department, suggesting that the yield of emergent CT scans in epileptic children with breakthrough seizures is low. This further corresponds with the recommendation by the American Academy of Neurology [19].

Children with chronic seizure disorders are likely to have been extensively evaluated by neurologists and undergone previous investigations such as magnetic resonance imaging [18]. Most patients (65%) in our study with a prior diagnosis of epilepsy did not have clinically significant findings on CT. An evidence based review looking at both adults and children recommended that emergency CT not be undertaken for patients with chronic seizures [19]. It is possible that in our cohort of patients, which included a large proportion of children with CSF shunts, a shunt malfunction may have presented with breakthrough seizures.

Acute meningitis may result in cerebral swelling and fatal herniation even without lumbar puncture [21]. In our study, the performance of CT to establish safety of LP in patients with suspected meningitis demonstrated a low yield of abnormal findings, with only two patients out of 70 noted to have radiological findings that preclude LP. This is consistent with the findings by Gopal et al., involving 113 adults, in which only 2.7% had absolute radiological contraindications to LP [16]. Other investigators demonstrated that normal head CT results do not guarantee safety of LP in children with suspected raised intracranial pressure especially in the setting of bacterial meningitis [20, 21]. An Australian study by Rennick, Shann and de Campo, looked at children with bacterial meningitis to assess whether the incidence of cerebral herniation increases immediately after lumbar puncture [21]. Normal CT results do not mean it is safe to perform a lumbar puncture in a paediatric patient with bacterial meningitis; clinical contraindications must not be ignored based on a normal CT result.

Normal head CT scans played a pivotal role in ruling out lesions and narrowing down the differential diagnoses. This made the emergency management of patients more efficient as the therapy was more targeted.

Relatively few decisions to scan were made by senior clinicians. It concerned us in this study that less than a third of decisions to do CT scan seem to have involved senior clinicians. This may be responsible for the poor screening of patients for CT revealed by our study. More senior input may be required before ordering scans. Over and above that, better clinical skills, especially checking for papilloedema and long tract signs, are vital in order to guide the scan requests and detect subtle pathology clinically.

Our study is limited by its retrospective design and the small sample size which did not give sufficient power to assess associations in a number of comparisons. Where univariate associations were noted, the small sample size precluded conducting of multivariable analysis to establish independent associations. In instances where some association has been established, the results must be interpreted with caution given the small sample size. As this is a single centre study, the findings may not be generalizable to all settings and must as such be interpreted with caution. There is a need to design properly powered studies to do more deeper analysis that look at the impact of CT head results on the clinical course of patients.

## Conclusion

Our study found that most children presenting acutely to the MEU have normal or clinically insignificant findings on CT. Patients with ventriculoperitoneal shunts had the highest yield of abnormal scans with hydrocephalus being the most common finding. Head CT has revolutionized the diagnosis and management of neurological illness in childhood, but possibly at the expense of good clinical skills and judgement. Thorough clinical assessment is still an indispensable and crucial tool in identifying patients that require CT brain e.g. presence of nausea or vomiting, papilloedema and long tract signs. Where CT is clearly indicated, the use of paediatric adjusted dose protocols, low dose scanning for follow-up scans and the potential for alternative non-radiation imaging such as MRI will also assist in reducing radiation risk.

## Acknowledgments

The authors would like to thank the staff in the Medical Records and Radiology departments at Red Cross War Memorial Children's Hospital, as well as all the patients included in the study from whom we all have much to learn every day.

## Author Contributions

**Conceptualization:** Rudzani Muloiwa.

**Formal analysis:** Rudzani Muloiwa.

**Supervision:** Heloise Buys, Tracy Kilborn, Rudzani Muloiwa.

**Writing – original draft:** Pamela Rudo Machingaidze.

**Writing – review & editing:** Heloise Buys, Tracy Kilborn, Rudzani Muloiwa.

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
