## [Decision Letter · Decision Letter 0]

24 Feb 2020

PONE-D-20-01190

Clinical use and indications for head computed tomography in children presenting with acute medical illness in a low- and middle-income setting

PLOS ONE

Dear Dr Buys,

Thank you for submitting your manuscript to PLOS ONE. After careful consideration, we feel that it has merit but does not fully meet PLOS ONE’s publication criteria as it currently stands. Therefore, we invite you to submit a revised version of the manuscript that addresses the points raised during the review process.

We would appreciate receiving your revised manuscript by Apr 09 2020 11:59PM. To enhance the reproducibility of your results, we recommend that if applicable you deposit your laboratory protocols in protocols.io, where a protocol can be assigned its own identifier (DOI) such that it can be cited independently in the future. For instructions see: http://journals.plos.org/plosone/s/submission-guidelines#loc-laboratory-protocols

A rebuttal letter that responds to each point raised by the academic editor and reviewer(s). The response should include a description of which changes were made and their location within the manuscript.  This letter should be uploaded as separate file and labeled 'Response to Reviewers'.A marked-up copy of your manuscript that highlights changes made to the original version. This file should be uploaded as separate file and labeled 'Revised Manuscript with Track Changes'.An unmarked version of your revised paper without tracked changes. This file should be uploaded as separate file and labeled 'Manuscript'.

We look forward to receiving your revised manuscript.

Kind regards,

Itamar Ashkenazi

Academic Editor

PLOS ONE

Journal Requirements:

Reviewers' comments:

Reviewer's Responses to Questions

**Comments to the Author**

1. Is the manuscript technically sound, and do the data support the conclusions?

Reviewer #1: Partly

Reviewer #2: Partly

Reviewer #3: Yes

2. Has the statistical analysis been performed appropriately and rigorously? 

Reviewer #1: No

Reviewer #2: Yes

Reviewer #3: I Don't Know

3. Have the authors made all data underlying the findings in their manuscript fully available?

Reviewer #1: No

Reviewer #2: No

Reviewer #3: Yes

4. Is the manuscript presented in an intelligible fashion and written in standard English?

Reviewer #1: Yes

Reviewer #2: Yes

Reviewer #3: Yes

5. Review Comments to the Author

Reviewer #1: The authors report a single center, retrospective, descriptive study of children who received a head CT within 24 hours of presentation to a hospital in Cape Town South Africa in 2013. They include 311 children with a median age of 39 months. They found the most common reason children had head CT imaging done was for seizure and altered mental status. They report abnormal CT findings were more common in patients with clinical manifestations, though I am unsure if the study was powered to detect these differences. They also report that children who had VP shunts were more likely to neurosurgical intervention after head CT. They conclude in the abstract, “Evidence-based institutional guidelines are warranted to ensure the best use of head- CT investigations in the management of patients.” However, they appear to be unfamiliar with the widely used and validated PECARN head trauma rule (see Kuppermann N, et al. Lancet. 2009. and the >100 articles that cite this original article).

The strengths of this study are the novelty of reporting on head CT use, results, and outcomes associated therein in an upper-middle-income country. This is an area in need of more work to further elucidate if patterns seen in high-income settings correspond to those in low- and middle-income countries.

Despite the article’s strengths, there are several weaknesses that this reviewer thinks should be addressed. Some of these weaknesses include the exclusion of trauma patients. Why exclude arguably the most common reason head CTs are ordered in some settings? It seems as though this is the case at this hospital too as the authors only include 311/2,837 head CTs done in a single year. Also, the authors should justify why a single year of data is reported, particularly since the data are somewhat old. The absence of clear justification for one year in a retrospective study raises concerns for selection bias. Is MRI available at this hospital or nearby? If so, this should be discussed and, at a minimum, the number of MRIs done should be mentioned as MRI has better resolution for assessing causes of seizure and for masses. Lastly, I was a little underwhelmed by the analysis in this paper. It is largely descriptive with little hypothesis testing, despite the study presenting a hypothesis that most CT results would be normal.

Abstract:

-It would be nice to have the overall rate of abnormal findings on head CT reported in the abstract as the authors report the breakdown of what the abnormalities were. It is currently unclear to me if the percentages reported for hydrocephalus (n=54, 57.4%) use the overall cohort as the denominator or not, it appears not.

-The line that reads, “Abnormal CT findings were commoner in patients with nausea or vomiting (n=21;9.3%, p=0.05)” is incorrect. The p-value is the table is >0.05.

Introduction:

-It’s unclear to me how the null hypothesis would be tested as the study appears to be descriptive in nature. Was this null hypothesis set a priori? If so, why did the authors feel this would be the case?

Methods:

-Why do the authors report just one year of data? I ask this in particular as 2013 is now 7 years ago. Was there a reason to ignore 2014-2019? Looking over this timeframe, or at least a longer timeframe than one year, would allow the authors to describe trends. This is particularly important as this is a single-center study with questionable generalizability.

-Does the line that reads, “Subjects were excluded if referral for CT was not done in the MEU as part of their assessment; injured children are seen in a separate trauma unit at this institution.” mean they excluded trauma patients? The abstract seems to allude to the same. If so, this needs to be justified as head trauma is a very common indication for head CT to evaluate for intracranial hemorrhage among children.

-Did the authors attempt to have a second radiologist review the CT scans for the article? If not, this should be justified or added to the Limitations. It is customary for radiology studies to have a second reviewer.

Results:

-Table 2 is difficult to read as formatted in the pdf. Please remove the overlapping line numbers from the text in the table.

-The results section reads as very long to me. The authors should consider more concise language and less repetition in the text of what is reported in the tables.

-Table 4 seems to be the most important table of the paper but seems buried after all of the merely descriptive data that precedes it. I do have concerns, however, about the reported significant findings. Was this study powered to detect these differences? A power calculation in the Methods would help with this. Also, again, using more years of data would allow the authors to reach more power and detect differences that may actually be present but the study is currently underpowered to detect.

-I suggest removing the section of the Results about the 17 month old who had a reaction to the contrast. This does not fit the focus of the manuscript.

-Why do they authors introduce LPs in the Results? This should at least be mentioned in the Methods if the authors choose to keep it in. My suggestion is to remove this from the manuscript or to abbreviate significantly as it detracts from the focus of the study (i.e. normal and abnormal CT scans).

Discussion:

-The Discussion seems to lack focus. I suggest the authors focus the Discussion on the 3-4 most compelling findings and not discuss every result with a comparison to extant literature.

-Why discuss odds for shunt revisions in patients with prior surgeries when the crux of this paper is CT indications, results, and outcomes from the CTs?

-The line that reads, “There was also a moderately strong association with nausea and vomiting.” is incorrect. The p-value is >0.05.

Limitations:

-This is the first mention of missing data. We should know how many variables were missing in each of the Tables.

-The authors should discuss why they only used one, outdated year of data in the Limitations or, even better, conduct the analysis with multiple years of data included.

Miscellaneous:

-The authors should be careful not to use abbreviations unless defined with the first appearance. Also, why use the abbreviations HCP and VPS in the Conclusions?

Under Data Availability, the authors state, “Data cannot be shared publicly because of this is a paediatric study.” To my knowledge, this should not preclude them from sharing de-identified data.

Reviewer #2: Thank you for the opportunity to review this paper.

This is an interesting retrospective paper reviewing the indications and outcomes of CT in children with acute medical illness.

Main issues:

• Although the data is interesting I think the focus should be different. From my perspective as a clinician the most relevant finding is in which cases the CT led to change in management.

• There are different indications for head CT. Analyzing this cases as a group may lead to misinterpretations and wrong management.

Minor issues:

• Introduction – Lines 70-71 (other effects of high-dose irradiation… ) are not relevant

• Methods- It is important to know if the hospital has clinical guidelines in which cases a CT should be ordered

Reviewer #3: PLOS One Peer Review

PONE-D-20-01190

February 21, 2020

Clinical use and indications for head computed tomography in children presenting with acute medical illness in a low- and middle-income setting

Overall Comments:

• This is an interesting study on CT scans for medical emergencies in a MLIC that has a fantastic database from which it can draw from. Overall the data as presented is expansive, but somewhat difficult to follow. Would suggest a stronger structure framed around clinical presentations (i.e. headache, seizure, etc) as these are usually quite clear with any patient that presents with a chief complaint.

o It is somewhat inappropriate to lump all positive CT results regardless of clinical presentation and analyze these as a whole (i.e. the clinical indication at baseline for a head ct win a patient with a VP shunt is significantly higher than for a headache…it is like comparing apples and oranges). This was done with table 1 and 3.

• Similarly, most of the data presented are primarily descriptive in nature. It seems as if there could be a bit more a deeper analysis of the rich data you have in addition to what has been down that would help a front line physician. For example:

o Emphasis on clinical features associated with positive CT heads for each clinical presentation

o Determining the impact of CT head results (either positive or negative) on the clinical course of patients

Abstract:

• Re results: would be useful to be presented with more data on what proportion of children with x clinical presentation (I.e. trauma, VP shunt concerns, headache) had positive findings, and what proportion required urgent neurosurgical intervention, rather than current cumulative data a presented for all patient. It is useful to know the clinical features associated with an abnormal CT scan as presented though. Adverse event data is hard to comment as the n’s are so small and does not seem to be a primary objective of the study. Would remove from abstract.

Introduction:

• Re primary objective of study (Pg4;Line80): This study is more of an observational study on CT use characteristics. It would be difficult to say it is to assess clinical utility, because it is retrospective, and not prospective. Would consider rewording.

• The stated null hypothesis (Pg 5;Line83-86) is awkward given your study design. If you have a null hypothesis would expect sample size and power calculations.

Methods:

• Pg 5;Line 98: The fact that this study only assessed MEDICAL EU and excluded Trauma related emergency indications is a significant limitation to the study design and pragmatic generalizability of the results. Trauma is a primary indication and utility for CT scanning children in an Emergency setting. Please indicate why trauma patients were not included and if feasible, would consider their inclusion in a reanalysis of the data. Many institutions in an LMIC may not have separate Medical and Trauma Emergency Units and see patients as a whole. Would consider explaining this rationale in either your introduction section more clearly.

• Re outcome measures/subgroup analysis, it would be interesting to assess practitioner adherence of practice to well established clinical decision tools for CT head (I.e. PECARN, CATCH or CHALICE).

• Minor point: Pg6;Line113: to be specific with terminology would change to “Pearson’s Chi-square test). Would also need to see sample size and power calculations as stated above.

Results:

• Table 1: would include demographic information broken down by clinical presentation (i.e. a column CT scans done for seizure, demographics of this group, etc.)

• Table 2: would revise to include additional data including: proportion with abnormal findings; proportion of neurosurgically significant CT head findings. Would also break down seizures between generalized and focal seizures.

• Consider potentially determining sensitivity and specificity of clinical presentation with CT findings (abnormal vs normal).

• Pg10;Lines196-200: Presented data is not very useful. This is mainly cumulative data and without context is hard to understand.

• Table 3: would be strengthened by including additional data columns on initial clinical presentation, and indications for CT head (clinical presentation). Could also include what proporition of CT findings were actually significant and resulted in a change in management

• Table 4: Although statistical signifance is achieved, it is difficult to know if this is valid given the likely inadequate sample size – would explicitly say this in the manuscript text somewhere (limitations?)

Management and Outcome:

• If one patient had a severe contrast reaction associated with CT head, cannot claim no adverse reactions occurred as stated abstract.

• This whole section is a bit oddly placed and not sure what it adds to the manuscript. Break down what proportion of patients got LPs based on clinical presentation.

• Is it possible for much of these data in this section to be summarized in a new table?

• Overall, results section would be strengthened and made more relevant if authors restructured the manuscript around clinical presentations (I.e. Meninigits/seizure/headache) and presented the data and clinical features for each clinical presentation (under clearly marked subtitled sections). Otherwise as presented, it is merely a lot of summative data with no context and therefore no relevance and little structure that makes it difficult for readers to follow.

Discussion:

• Structure of discussion is overall much easier to follow as it is based around clinical presentations. Would still benefit from more structure as stated above.

• Pg18;line 379: I am likewise concerned with a number of statements about significance when clinical sample size is not determined. It deserves more attention rather than a tag-along sentence mentioning missing data. A major point that should also be discussed is the omission of trauma patients (a primary driver of head CT usage in an ED).

• Likewise, I would think a major limitation is your study design – it is an observational study with no overt comparator group.

• Pg18l;line 379: Would clearly indicate this is a limitations section.

Minor point throughout manuscript:

• “Commonest” is awkward and seems a bit too colloquial [albeit a word still]. Consider “most common”. Likewise, with “commoner” and “more common”.

6. PLOS authors have the option to publish the peer review history of their article (what does this mean?). If published, this will include your full peer review and any attached files.

Reviewer #1: No

Reviewer #2: No

Reviewer #3: No

---

## [Author Response · Author response to Decision Letter 0]

25 Aug 2020

Reviewer 1

 Comment# Reviewer Comment Response to review comment

 1 The authors report a single center, retrospective, descriptive study of children who received a head CT within 24 hours of presentation to a hospital in Cape Town South Africa in 2013. They include 311 children with a median age of 39 months. They found the most common reason children had head CT imaging done was for seizure and altered mental status. They report abnormal CT findings were more common in patients with clinical manifestations, though I am unsure if the study was powered to detect these differences. They also report that children who had VP shunts were more likely to neurosurgical intervention after head CT. They conclude in the abstract, “Evidence-based institutional guidelines are warranted to ensure the best use of head- CT investigations in the management of patients.” However, they appear to be unfamiliar with the widely used and validated PECARN head trauma rule (see Kuppermann N, et al. Lancet. 2009. and the >100 articles that cite this original article). Thank you for this feedback. We are well aware of the PECARN head trauma rule, but it was not applicable to this study. None of the patients reviewed here underwent head CT for trauma related reasons. At this hospital there is a separate trauma unit which is not linked to the Medical Emergency Unit and indications for computed tomography (CT) in the trauma unit are according to the trauma guidelines. Patients with hydrocephalus and ventriculoperitoneal shunts are evaluated in the Medical Emergency Unit, hence they were included in this study, as shunt complications may also present in a manner similar to medical conditions. Once again, trauma patients were not included in this study as the guidelines for imaging these patients are already set according to international norms (including a PECARN based approach). We have added the following section to make this clear, 

“At this hospital there is a separate trauma unit which is not linked to the medical emergency unit (MEU). Indications for computed tomography (CT) and subsequent management in the trauma unit follow well established international trauma guidelines. Although the hospital has clinical guidelines for performing CT in the MEU, these are relatively flexible and there are frequent deviations from these guidelines.” Lines 91-95

 2 The strengths of this study are the novelty of reporting on head CT use, results, and outcomes associated therein in an upper-middle-income country. This is an area in need of more work to further elucidate if patterns seen in high-income settings correspond to those in low- and middle-income countries Thank you for the comment. 

 3 Despite the article’s strengths, there are several weaknesses that this reviewer thinks should be addressed. Some of these weaknesses include the exclusion of trauma patients. Why exclude arguably the most common reason head CTs are ordered in some settings? It seems as though this is the case at this hospital too as the authors only include 311/2,837 head CTs done in a single year. Also, the authors should justify why a single year of data is reported, particularly since the data are somewhat old. The absence of clear justification for one year in a retrospective study raises concerns for selection bias. Is MRI available at this hospital or nearby? If so, this should be discussed and, at a minimum, the number of MRIs done should be mentioned as MRI has better resolution for assessing causes of seizure and for masses. Lastly, I was a little underwhelmed by the analysis in this paper. It is largely descriptive with little hypothesis testing, despite the study presenting a hypothesis that most CT results would be normal. Thank you for raising this point. Trauma patients were not included in this study as the guidelines for imaging these patients are already set according to international norms. The study was designed to focus specifically on head CT scans done for medical indications as this is a grey area we felt needed to be addressed. 

According to the sample size calculation a period of 1 year would yield a sufficient sample size with meaningful precision without a need to have to decide on sampling framework. As all participants that made inclusion criteria were enrolled, this design was chosen precisely in order to avoid selection bias. In our estimate, adding more years (and therefore participants) would increase the cost of the study but without obvious benefit. Given our limited resources, we regard the issue of sample size as an ethical issue. We have added a section to clarify this that reads, 

“We estimated that over a one-year period about 250 to 350 children seen in the medical emergency unit would make inclusion criteria. Based on this sample size, for outcomes ranging from 5% to 20%, the precision will fall within 3% of the point estimate.” Lines 115-118

The hospital is in a relatively resource-limited setting. Magnetic resonance imaging (MRI) is available but access to it strictly limited and not as readily as CT. In addition, MRI requires an anaesthetist and must be scheduled in advance, whereas CT may only require light sedation and is available almost immediately in an emergency setting.

The study was designed mainly to describe CT findings in a non-trauma setting in children, with hypothesis testing only a secondary outcome. For most of the analysis we did, descriptive statistics indicated no signals of difference between groups. Where there seem to be a difference, we went ahead and did hypothesis testing as shown in Table 4. We considered doing advanced modelling but decided against this as unjustified ‘torturing’ of data that may end up creating associations that we ourselves may have created.

 4 Abstract:

-It would be nice to have the overall rate of abnormal findings on head CT reported in the abstract as the authors report the breakdown of what the abnormalities were. It is currently unclear to me if the percentages reported for hydrocephalus (n=54, 57.4%) use the overall cohort as the denominator or not, it appears not.

-The line that reads, “Abnormal CT findings were commoner in patients with nausea or vomiting (n=21;9.3%, p=0.05)” is incorrect. The p-value is the table is >0.05. The entire abstract has been revised to take into account the revisions made in the manuscript as well as to take into account the comments made specifically about the abstract here. Lines 24-50

-Thank you, this has been removed with revision of the abstract in response to other reviewer comments.

 5 Introduction:

-It’s unclear to me how the null hypothesis would be tested as the study appears to be descriptive in nature. Was this null hypothesis set a priori? If so, why did the authors feel this would be the case Thank you for pointing this out. The primary aim of the study was indeed descriptive. The ‘hypothesis’ here refers to the secondary outcome. As this was opportunistic a sample size for testing this was never calculated. To remove confusion this sentence has been removed and the stating of the secondary outcome rephrased to now read, “Secondarily, we compare frequencies of presenting factors between participants with normal and abnormal findings on CT.” Lines 82-83

 6 Methods:

-Why do the authors report just one year of data? I ask this in particular as 2013 is now 7 years ago. Was there a reason to ignore 2014-2019? Looking over this timeframe, or at least a longer timeframe than one year, would allow the authors to describe trends. This is particularly important as this is a single-center study with questionable generalizability. The issue of the choice of a single year has already been addressed. While we agree that the data seem somewhat old, we feel that the issue the study addresses remain pertinent and that the data are not outdated as such and are able to add value to this area. 

In order to reduce risk of overinterpreting results, we have added the following sentence to the Discussion - “As this is a single centre study, the findings may not be generalizable to all settings and must as such be interpreted with caution.” Lines 312-316

 7 -Does the line that reads, “Subjects were excluded if referral for CT was not done in the MEU as part of their assessment; injured children are seen in a separate trauma unit at this institution.” mean they excluded trauma patients? The abstract seems to allude to the same. If so, this needs to be justified as head trauma is a very common indication for head CT to evaluate for intracranial hemorrhage among children. We have already dealt with the issue of excluding trauma patients above. 

 8 -Did the authors attempt to have a second radiologist review the CT scans for the article? If not, this should be justified or added to the Limitations. It is customary for radiology studies to have a second reviewer Thank you for you for pointing this out. It is the practice in the hospital to have the CT scans reported by a junior radiologist and then reviewed by a senior radiologist. We have clarified this in methods with a section that reads,

 “CT findings as independently reported by a junior radiologist and reviewed by an experienced paediatric radiologist were noted.” Lines 101-104

 9 Results:

-Table 2 is difficult to read as formatted in the pdf. Please remove the overlapping line numbers from the text in the table.

 Thank you for pointing this out. The table has been rechecked for clarity.

 10 -The results section reads as very long to me. The authors should consider more concise language and less repetition in the text of what is reported in the tables.

 Thank you for pointing this out. Sections have been modified in part to respond to other review comments. We hope this has helped to focus the results section.

 11 -Table 4 seems to be the most important table of the paper but seems buried after all of the merely descriptive data that precedes it. I do have concerns, however, about the reported significant findings. Was this study powered to detect these differences? A power calculation in the Methods would help with this. Also, again, using more years of data would allow the authors to reach more power and detect differences that may actually be present but the study is currently underpowered to detect. As previously mentioned, this was a secondary opportunistic outcome for which the study was never powered. We have now included in our methods the basis for our current sample size, that read “ We estimated that over a one year period about 250 to 350 children seen in the medical emergency unit would make inclusion criteria. Based on this sample size, for outcomes ranging from 5% to 20%, the precision will fall within 3% of the point estimate.” Lines 115-118 

We had of course hoped that this would generate viable hypothesis that would need follow up studies specifically designed to answer such hypothesis.

We have noted this as a limitation of our study. “Where univariate associations were noted, the small sample size precluded conducting of multivariable analysis to establish independent associations. In instances where some association has been established, the results must be interpreted with caution given the small sample size.” Lines 310-313

 12 -I suggest removing the section of the Results about the 17 month old who had a reaction to the contrast. This does not fit the focus of the manuscript.

 The section regarding the 17-month-old who experienced a severe contrast reaction has been removed.

 13 -Why do they authors introduce LPs in the Results? This should at least be mentioned in the Methods if the authors choose to keep it in. My suggestion is to remove this from the manuscript or to abbreviate significantly as it detracts from the focus of the study (i.e. normal and abnormal CT scans) Thank you for your suggestion. Lumbar punctures were mentioned as some of the indications for doing a head CT were to establish whether there were any radiological contraindications to performing a lumbar puncture. The section has been abbreviated in order to focus the paper.

 14 Discussion:

-The Discussion seems to lack focus. I suggest the authors focus the Discussion on the 3-4 most compelling findings and not discuss every result with a comparison to extant literature.

-Why discuss odds for shunt revisions in patients with prior surgeries when the crux of this paper is CT indications, results, and outcomes from the CTs?

-The line that reads, “There was also a moderately strong association with nausea and vomiting.” is incorrect. The p-value is >0.05. 

• The discussion has been revised to give it better focus.

• We thought it pertinent to mention this as head CT is a crucial tool in the investigation of possible shunt malfunction and subsequent revision, which so happens to be more common in children compared to adults.

• Thank you once again for highlighting this, the line has been removed.

 15 Limitations:

-This is the first mention of missing data. We should know how many variables were missing in each of the Tables.

-The authors should discuss why they only used one, outdated year of data in the Limitations or, even better, conduct the analysis with multiple years of data included • The missing data related to lumbar punctures, specifically with respect to opening pressures and indications for measuring this in some and not others. As we have removed much of the data on lumbar punctures as per the response to an earlier comment, this no longer applies and the reference to missing data has been removed.

• The year 2013 was the most recent year with complete data at the time the study was conceptualized. As already indicated, the data from that single year was deemed sufficient to give the requisite sample size to answer the primary question at an affordable rate without causing selection bias (as all qualifying participants would be included.) Although it can be reasonably argued that the data are ‘old’ they are not ‘outdated’ as far their relevance is concerned. The practice of using CT scan for non-trauma medical emergency remains both in our hospital as well as in large number of low- and middle-income settings as a result of lack of access to MRI that we mention in the manuscript.

 16 Miscellaneous:

-The authors should be careful not to use abbreviations unless defined with the first appearance. Also, why use the abbreviations HCP and VPS in the Conclusions? 

-We have noted this. The manuscript has been reviewed and amended to deal with this issue

 17 Under Data Availability, the authors state, “Data cannot be shared publicly because of this is a paediatric study.” To my knowledge, this should not preclude them from sharing de-identified data. This was a misunderstanding on our part. We had thought that the request was for raw data for which IRB had not given consent. All data that meet Plos’s minimal dataset have been presented in the article, including those used in any revisions.

Reviewer 2

 Comment# Review comment Response to review comment

 1 Although the data is interesting I think the focus should be different. From my perspective as a clinician the most relevant finding is in which cases the CT led to change in management. Thank you for the observation. We agree with your perspective and highlighting which patients received surgical intervention post CT scan was one of the aims of the study. In the Introduction we state, “ Our study aimed to describe the clinical use of emergency head CT scans at a tertiary paediatric hospital in a low and middle-income country setting. The primary outcomes of interest were indications for and findings of head CT imaging in children presenting for acute medical care, as well as to establish baseline characteristics and interventions performed post CT scanning.” Lines 78-82

 2 There are different indications for head CT. Analyzing these cases as a group may lead to misinterpretations and wrong management

 Thank you. While we acknowledge that there are different head CT indications, the indications have been stratified to highlight these differences. We have additionally added extra columns to Tables 2 and 3 to further stratify the data such that groups with different characters are analyzed separately.

 3 Introduction – Lines 70-71 (other effects of high-dose irradiation …) are not relevant Thank you for the input. The sentence has been removed.

 4 Methods- It is important to know if the hospital has clinical guidelines in which cases a CT should be ordered The hospital does have clinical guidelines, however these are relatively flexible. Head CT scan requests are still discussed on a patient by patient basis as they are not set in stone. We have added the following sentence to clarify this-

“Although the hospital has clinical guidelines for performing CT in the MEU, these are relatively flexible and there are frequent deviations from these guidelines.” Lines 94-95

Reviewer 3

 Comment# Review comment Response to review comment

 1 This is an interesting study on CT scans for medical emergencies in a MLIC that has a fantastic database from which it can draw from. Overall the data as presented is expansive, but somewhat difficult to follow. Would suggest a stronger structure framed around clinical presentations (i.e. headache, seizure, etc) as these are usually quite clear with any patient that presents with a chief complaint Thank you for the positive feedback, it is much appreciated. Although we agree that structuring the analysis around clinical presentation would make for a rational structure, we found it more meaningful to use indications for CT scans as the starting place for our analysis. The reason for this is that it is not so much the clinical presentation that dictates whether a request for CT would be granted or not, but rather justification of why such a request is being made. That is, for each request, a justification (indication) is demanded. Often the indication combined multiple clinical features.

 2 It is somewhat inappropriate to lump all positive CT results regardless of clinical presentation and analyze these as a whole (i.e. the clinical indication at baseline for a head ct win a patient with a VP shunt is significantly higher than for a headache…it is like comparing apples and oranges). This was done with table 1 and 3. We acknowledge this input. Table 1 was intended to show the baseline characteristics of the group with no analysis done. We have updated Table 3 to show the VP shunt group on its own.

 3 Similarly, most of the data presented are primarily descriptive in nature. It seems as if there could be a bit more a deeper analysis of the rich data you have in addition to what has been down that would help a front line physician. For example:

o Emphasis on clinical features associated with positive CT heads for each clinical presentation

o Determining the impact of CT head results (either positive or negative) on the clinical course of patients We partly responded to this question above. The study was designed as descriptive study and not specifically designed with the requisite power for a primary hypothesis testing design. As already pointed out in our response to this reviewer, even where ‘statistical significance’ is found, the results must be interpreted with caution. We agree fully with this review perspective and believe we should in the current manuscript avoid over-analysis of the data that risks generating spurious results, both positive and negative. We have however added additional analysis as shown in Table 3 and associated text that reads, 

“ There were 29 (46.8% ) of the 62 patients with VPS that had at least one abnormal finding on CT (interval change) compared to 63 (25.3%) of the 249 without VPS (i.e. no new findings or interval change if previously scanned); P=0.001.” (Lines 185-187).

 We have also added the following sentence to the Discussion section to highlight the need of studies designed specifically for hypothesis testing, “There is a need to design properly powered studies to do more deeper analysis that look at the impact of CT head results on the clinical course of patients.” Lines 314-316

 4 Abstract:

• Re results: would be useful to be presented with more data on what proportion of children with x clinical presentation (I.e. trauma, VP shunt concerns, headache) had positive findings, and what proportion required urgent neurosurgical intervention, rather than current cumulative data a presented for all patient. It is useful to know the clinical features associated with an abnormal CT scan as presented though. Adverse event data is hard to comment as the n’s are so small and does not seem to be a primary objective of the study. Would remove from abstract -Thank you for this fair criticism. The abstract has undergone a major revision to take into consideration all the issues raised including showing the data as suggested.

 5 Introduction:

• Re primary objective of study (Pg4;Line80): This study is more of an observational study on CT use characteristics. It would be difficult to say it is to assess clinical utility, because it is retrospective, and not prospective. Would consider rewording.

• The stated null hypothesis (Pg 5;Line83-86) is awkward given your study design. If you have a null hypothesis would expect sample size and power calculations 

• The primary aim of the study was indeed descriptive. The sentence has been modified to say, 

 “Our study aimed to describe the clinical use of emergency head CT scans at a tertiary paediatric hospital in a low- and middle-income country (LMIC) setting.” Lines 78-79.

• ‘Hypothesis’ here refers to ‘assumptions’ made, rather than hypothesis testing per se. To remove confusion this sentence has been removed and the stating of the secondary outcome rephrased to now read,

 “Secondarily, we compare frequencies of presenting factors between participants with normal and abnormal findings on CT.” Lines 82-83

 6 Methods:

• Pg 5;Line 98: The fact that this study only assessed MEDICAL EU and excluded Trauma related emergency indications is a significant limitation to the study design and pragmatic generalizability of the results. Trauma is a primary indication and utility for CT scanning children in an Emergency setting. Please indicate why trauma patients were not included and if feasible, would consider their inclusion in a reanalysis of the data. Many institutions in an LMIC may not have separate Medical and Trauma Emergency Units and see patients as a whole. Would consider explaining this rationale in either your introduction section more clearly.

• Re outcome measures/subgroup analysis, it would be interesting to assess practitioner adherence of practice to well established clinical decision tools for CT head (I.e. PECARN, CATCH or CHALICE).

• Minor point: Pg6;Line113: to be specific with terminology would change to “Pearson’s Chi-square test). Would also need to see sample size and power calculations as stated above. • Thank you for raising this point. At this hospital there is a separate trauma unit which is not linked to the Medical Emergency Unit and indications for computed tomography (CT) in the trauma unit are according to the trauma guidelines. Trauma patients were not included in this study as the guidelines for imaging these patients are already set according to international norms. The study was designed to focus specifically on head CT scans done for medical indications as this is a grey area we felt needed to be addressed in this institution. PECARN, CATCH and CHALICE are all applicable to head trauma which was not the focus of the study as we set out to analyse the indications in a medical setting. The following section has been added to the introduction to explain the rationale, 

 “While there are clear guidelines for CT following head trauma, the same is not the case for CT head in non-trauma medical emergency.[9, 10] This means that requests for CT scan are generally made on unclear bases which has significant implications when there are significant resource limitations such as in low and middle income country (LMIC) settings.” Lines 71-74. Two additional references have been added to the section:

9. Khalifa M, Gallego B. Grading and assessment of clinical predictive tools for paediatric head injury: a new evidence-based approach. BMC Emerg Med. 2019;19(1):35. Epub 2019/06/16. doi: 10.1186/s12873-019-0249-y. PubMed PMID: 31200643; PubMed Central PMCID: PMCPMC6570950.

10. Lyttle MD, Crowe L, Oakley E, Dunning J, Babl FE. Comparing CATCH, CHALICE and PECARN clinical decision rules for paediatric head injuries. Emerg Med J. 2012;29(10):785-94. Epub 2012/02/01. doi: 10.1136/emermed-2011-200225. PubMed PMID: 22291431.

Lines 359-365

• As explained above, there are no clear decision tools currently in use outside trauma settings. We did not think it appropriate to use head trauma clinical decision tools in this context.

• The amendment to the terminology has been made. The sample size calculation has been included. The following lines have now been added to the Methods section, 

“Categorical variables were compared using Pearson’s Chi-square or Fisher’s exact tests as appropriate. We estimated that over a one-year period about 250 to 350 children seen in the medical emergency unit would make inclusion criteria. Based on this sample size, for outcomes ranging from 5% to 20%, the precision will fall within 3% of the point estimate.” Lines 114-118

 7 Results:

• Table 1: would include demographic information broken down by clinical presentation (i.e. a column CT scans done for seizure, demographics of this group, etc.)

• Table 2: would revise to include additional data including: proportion with abnormal findings; proportion of neurosurgically significant CT head findings. Would also break down seizures between generalized and focal seizures.

• Consider potentially determining sensitivity and specificity of clinical presentation with CT findings (abnormal vs normal).

• Pg10; Lines196-200: Presented data is not very useful. This is mainly cumulative data and without context is hard to understand.

• Table 3: would be strengthened by including additional data columns on initial clinical presentation, and indications for CT head (clinical presentation). Could also include what proportion of CT findings were actually significant and resulted in a change in management

• Table 4: Although statistical signifance is achieved, it is difficult to know if this is valid given the likely inadequate sample size – would explicitly say this in the manuscript text somewhere (limitations? 

• Table 1 shows baseline characteristics of the whole group. We believe that this should remain as is in order for the reader to understand the sampled population. We do not believe that stratifying on the bases of the listed variables adds any extra value. But we agree that stratification as mentioned previously is necessary to bring clarity to other parts of the manuscript and have as a consequence created an extra column in Table 3 as indicated previously.

• We have added data on abnormal findings to Table 2 but decided against adding the other things to the table as this made the table cumbersome and confusing. Data on neurosurgically significant CT head findings and seizure types are contained in text (Lines 225-227 and Lines 165-167, respectively) as well as (new) Table 5.

• While the idea of sensitivity and specificity is quite attractive to us, we fear that it communicates something that we are not sure we can sustain. A normal finding on CT will by itself not indicate that everything is normal in the presence of clinical illness. While proportions of abnormal CT’s with specific clinical signs may approximate the sensitivity of the clinical sign, absence of the same signs as well as absence of findings on CT cannot be interpreted to estimate specificity.

• We felt that it would be useful to show aggregate data as an introduction to this section. We have revised this section slightly to present it in a way we hope is both succinct and more useful.

• We have updated Table 3 by stratifying the data to add a column of those with VP shunts as suggested by a previous comment. We found that attempting to add more data to this table made the table cumbersome and confusing. The other data requested is contained elsewhere in the manuscript such as the new Table 5. 

• We have included the following line to the section indicating limitations of the study, “In instances where some association has been established, the results must be interpreted with caution given the small sample size.” Lines 312-313

 8 Management and Outcome:

• If one patient had a severe contrast reaction associated with CT head, cannot claim no adverse reactions occurred as stated abstract.

• This whole section is a bit oddly placed and not sure what it adds to the manuscript. Break down what proportion of patients got LPs based on clinical presentation.

• Is it possible for much of these data in this section to be summarized in a new table?

• Overall, results section would be strengthened and made more relevant if authors restructured the manuscript around clinical presentations (I.e. Meningitis/seizure/headache) and presented the data and clinical features for each clinical presentation (under clearly marked subtitled sections). Otherwise as presented, it is merely a lot of summative data with no context and therefore no relevance and little structure that makes it difficult for readers to follow. 

• Mention of this case has been removed in response to a previous comment that suggested that this may be a distraction to the message of the manuscript.

• The section on lumbar puncture has been largely revised and made succinct to focus it and stop it distracting from the theme of the manuscript. Lines 222 to 223 

• Table 5 has been created to accommodate some of the data.

• We have responded to the approach we took in reporting the result in a previous response. (Reviewer 3, comment 1). More analysis has been added as shown in Table 3 and related text. Lines 185-187. 

 9 Discussion:

• Structure of discussion is overall much easier to follow as it is based around clinical presentations. Would still benefit from more structure as stated above.

• Pg18;line 379: I am likewise concerned with a number of statements about significance when clinical sample size is not determined. It deserves more attention rather than a tag-along sentence mentioning missing data. A major point that should also be discussed is the omission of trauma patients (a primary driver of head CT usage in an ED).

• Likewise, I would think a major limitation is your study design – it is an observational study with no overt comparator group.

• Pg18l;line 379: Would clearly indicate this is a limitations section. Thank you for the affirmative feedback. 

• We have already responded above to the suggestion of having a clinical presentation based structure. In general, the discussion has seen massive revisions in response to review comments that we hope gives it a better and clearer structure.

• We have responded to all the issues raised here elsewhere in our responses

• We agree that retrospective studies are fraught with limitations which we have noted as such as we noted, “Our study is limited by its retrospective design ...” Line 309. However, we also believe that such studies provide ‘low hanging fruit’ especially in poor resourced areas to generate hypothesis and to audit practice which others can learn from at a relatively affordable rate.

 10 Minor point throughout manuscript:

• “Commonest” is awkward and seems a bit too colloquial [albeit a word still]. Consider “most common”. Likewise, with “commoner” and “more common” Thank you for the feedback.

“Commonest” and commoner have been respectively replaced with “Most common” and “more common” throughout the manuscript.

---

## [Editor Report · Decision Letter 1]

1 Sep 2020

PONE-D-20-01190R1

Clinical use and indications for head computed tomography in children presenting with acute medical illness in a low- and middle-income setting

PLOS ONE

Dear Dr. Buys,

I found it very difficult to follow the revision you submitted. Your comments and the reviewers' comments are joined together.  Please rewrite your cover letter placing spaces between the reviewers' comments and your responses. Within your responses please place spaces between your discussion and the description of the changes that were made.

In your marked copy you highlighted the changes that were made. However, we cannot appreciate these changes if we cannot see those paragraphs, sentences and words that were erased. Please resubmit a marked copy in which changes can be easily identified: both those words/sentences/paragraphs you erased, and those that were added.

Only then I will resend your revised mansucript to the reviewers.

Sincerely,

Itamar Ashkenazi

Academic Editor

PLOS ONE

---

## [Author Response · Author response to Decision Letter 1]

5 Sep 2020

Review comments to the authors with author responses

Reviewer 1

The authors report a single center, retrospective, descriptive study of children who received a head CT within 24 hours of presentation to a hospital in Cape Town South Africa in 2013. They include 311 children with a median age of 39 months. They found the most common reason children had head CT imaging done was for seizure and altered mental status. They report abnormal CT findings were more common in patients with clinical manifestations, though I am unsure if the study was powered to detect these differences. They also report that children who had VP shunts were more likely to neurosurgical intervention after head CT. They conclude in the abstract, “Evidence-based institutional guidelines are warranted to ensure the best use of head- CT investigations in the management of patients.” However, they appear to be unfamiliar with the widely used and validated PECARN head trauma rule (see Kuppermann N, et al. Lancet. 2009. and the >100 articles that cite this original article).

Thank you for this feedback. We are well aware of the PECARN head trauma rule, but it was not applicable to this study. None of the patients reviewed here underwent head CT for trauma related reasons. At this hospital there is a separate trauma unit which is not linked to the Medical Emergency Unit and indications for computed tomography (CT) in the trauma unit are according to the trauma guidelines. Patients with hydrocephalus and ventriculoperitoneal shunts are evaluated in the Medical Emergency Unit, hence they were included in this study, as shunt complications may also present in a manner similar to medical conditions. Once again, trauma patients were not included in this study as the guidelines for imaging these patients are already set according to international norms (including a PECARN based approach). We have added the following section to make this clear, 

“At this hospital there is a separate trauma unit which is not linked to the medical emergency unit (MEU). Indications for computed tomography (CT) and subsequent management in the trauma unit follow well established international trauma guidelines. Although the hospital has clinical guidelines for performing CT in the MEU, these are relatively flexible and there are frequent deviations from these guidelines.” Lines 93-97

The strengths of this study are the novelty of reporting on head CT use, results, and outcomes associated therein in an upper-middle-income country. This is an area in need of more work to further elucidate if patterns seen in high-income settings correspond to those in low- and middle-income countries

Thank you for the comment.

Despite the article’s strengths, there are several weaknesses that this reviewer thinks should be addressed. Some of these weaknesses include the exclusion of trauma patients. Why exclude arguably the most common reason head CTs are ordered in some settings? It seems as though this is the case at this hospital too as the authors only include 311/2,837 head CTs done in a single year. Also, the authors should justify why a single year of data is reported, particularly since the data are somewhat old. The absence of clear justification for one year in a retrospective study raises concerns for selection bias. Is MRI available at this hospital or nearby? If so, this should be discussed and, at a minimum, the number of MRIs done should be mentioned as MRI has better resolution for assessing causes of seizure and for masses. Lastly, I was a little underwhelmed by the analysis in this paper. It is largely descriptive with little hypothesis testing, despite the study presenting a hypothesis that most CT results would be normal.

Thank you for raising this point. Trauma patients were not included in this study as the guidelines for imaging these patients are already set according to international norms. The study was designed to focus specifically on head CT scans done for medical indications as this is a grey area we felt needed to be addressed. 

According to the sample size calculation a period of 1 year would yield a sufficient sample size with meaningful precision without a need to have to decide on sampling framework. As all participants that made inclusion criteria were enrolled, this design was chosen precisely in order to avoid selection bias. In our estimate, adding more years (and therefore participants) would increase the cost of the study but without obvious benefit. Given our limited resources, we regard the issue of sample size as an ethical issue. We have added a section to clarify this that reads, 

“We estimated that over a one-year period about 250 to 350 children seen in the medical emergency unit would make inclusion criteria. Based on this sample size, for outcomes ranging from 5% to 20%, the precision will fall within 3% of the point estimate.” Lines 117-120

The hospital is in a relatively resource-limited setting. Magnetic resonance imaging (MRI) is available but access to it strictly limited and not as readily as CT. In addition, MRI requires an anaesthetist and must be scheduled in advance, whereas CT may only require light sedation and is available almost immediately in an emergency setting.

The study was designed mainly to describe CT findings in a non-trauma setting in children, with hypothesis testing only a secondary outcome. For most of the analysis we did, descriptive statistics indicated no signals of difference between groups. Where there seem to be a difference, we went ahead and did hypothesis testing as shown in Table 4. We considered doing advanced modelling but decided against this as unjustified ‘torturing’ of data that may end up creating associations that we ourselves may have created.

ABSTRACT:

-It would be nice to have the overall rate of abnormal findings on head CT reported in the abstract as the authors report the breakdown of what the abnormalities were. It is currently unclear to me if the percentages reported for hydrocephalus (n=54, 57.4%) use the overall cohort as the denominator or not, it appears not.

The entire abstract has been revised to take into account the revisions made in the manuscript as well as to take into account the comments made specifically about the abstract here. Lines 26-52

-The line that reads, “Abnormal CT findings were commoner in patients with nausea or vomiting (n=21;9.3%, p=0.05)” is incorrect. The p-value is the table is >0.05.

Thank you, this has been removed with complete revision of the abstract in response to other reviewer comments.

INTRODUCTION:

-It’s unclear to me how the null hypothesis would be tested as the study appears to be descriptive in nature. Was this null hypothesis set a priori? If so, why did the authors feel this would be the case

Thank you for pointing this out. The primary aim of the study was indeed descriptive. The ‘hypothesis’ here refers to the secondary outcome. As this was opportunistic a sample size for testing this was never calculated. To remove confusion this sentence has been removed and the stating of the secondary outcome rephrased to now read, “Secondarily, we compare frequencies of presenting factors between participants with normal and abnormal findings on CT.” Lines 84-85

METHODS:

-Why do the authors report just one year of data? I ask this in particular as 2013 is now 7 years ago. Was there a reason to ignore 2014-2019? Looking over this timeframe, or at least a longer timeframe than one year, would allow the authors to describe trends. This is particularly important as this is a single-center study with questionable generalizability.

The issue of the choice of a single year has already been addressed. While we agree that the data seem somewhat old, we feel that the issue the study addresses remain pertinent and that the data are not outdated as such and are able to add value to this area. 

In order to reduce risk of overinterpreting results, we have added the following sentence to the Discussion - “As this is a single centre study, the findings may not be generalizable to all settings and must as such be interpreted with caution.” Lines 319-320

-Does the line that reads, “Subjects were excluded if referral for CT was not done in the MEU as part of their assessment; injured children are seen in a separate trauma unit at this institution.” mean they excluded trauma patients? The abstract seems to allude to the same. If so, this needs to be justified as head trauma is a very common indication for head CT to evaluate for intracranial hemorrhage among children.

We have already dealt with the issue of excluding trauma patients above.

-Did the authors attempt to have a second radiologist review the CT scans for the article? If not, this should be justified or added to the Limitations. It is customary for radiology studies to have a second reviewer

Thank you for you for pointing this out. It is the practice in the hospital to have the CT scans reported by a junior radiologist and then reviewed by a senior radiologist. We have clarified this in methods with a section that reads,

 “CT findings as independently reported by a junior radiologist and reviewed by an experienced paediatric radiologist were noted.” Lines 103-105

RESULTS:

-Table 2 is difficult to read as formatted in the pdf. Please remove the overlapping line numbers from the text in the table.

Thank you for pointing this out. The table has been rechecked for clarity.

-The results section reads as very long to me. The authors should consider more concise language and less repetition in the text of what is reported in the tables.

Thank you for pointing this out. Sections have been modified in part to respond to other review comments. We hope this has helped to focus the results section.

-Table 4 seems to be the most important table of the paper but seems buried after all of the merely descriptive data that precedes it. I do have concerns, however, about the reported significant findings. Was this study powered to detect these differences? A power calculation in the Methods would help with this. Also, again, using more years of data would allow the authors to reach more power and detect differences that may actually be present but the study is currently underpowered to detect.

As previously mentioned, this was a secondary opportunistic outcome for which the study was never powered. We have now included in our methods the basis for our current sample size, that read “ We estimated that over a one year period about 250 to 350 children seen in the medical emergency unit would make inclusion criteria. Based on this sample size, for outcomes ranging from 5% to 20%, the precision will fall within 3% of the point estimate.” Lines 117-120 

We had of course hoped that this would generate viable hypothesis that would need follow up studies specifically designed to answer such hypothesis.

We have noted this as a limitation of our study. “Where univariate associations were noted, the small sample size precluded conducting of multivariable analysis to establish independent associations. In instances where some association has been established, the results must be interpreted with caution given the small sample size.” Lines 316-319

-I suggest removing the section of the Results about the 17-month-old who had a reaction to the contrast. This does not fit the focus of the manuscript.

The section regarding the 17-month-old who experienced a severe contrast reaction has been removed.

-Why do they authors introduce LPs in the Results? This should at least be mentioned in the Methods if the authors choose to keep it in. My suggestion is to remove this from the manuscript or to abbreviate significantly as it detracts from the focus of the study (i.e. normal and abnormal CT scans)

Thank you for your suggestion. Lumbar punctures were mentioned as some of the indications for doing a head CT were to establish whether there were any radiological contraindications to performing a lumbar puncture. The section has been abbreviated in order to focus the paper.

DISCUSSION:

-The Discussion seems to lack focus. I suggest the authors focus the Discussion on the 3-4 most compelling findings and not discuss every result with a comparison to extant literature.

The discussion has been revised to give it better focus.

-Why discuss odds for shunt revisions in patients with prior surgeries when the crux of this paper is CT indications, results, and outcomes from the CTs?

We thought it pertinent to mention this as head CT is a crucial tool in the investigation of possible shunt malfunction and subsequent revision, which so happens to be more common in children compared to adults.

-The line that reads, “There was also a moderately strong association with nausea and vomiting.” is incorrect. The p-value is >0.05.

Thank you once again for highlighting this, the line has been removed.

LIMITATIONS

-This is the first mention of missing data. We should know how many variables were missing in each of the Tables.

The missing data related to lumbar punctures, specifically with respect to opening pressures and indications for measuring this in some and not others. As we have removed much of the data on lumbar punctures as per the response to an earlier comment, this no longer applies and the reference to missing data has been removed.

-The authors should discuss why they only used one, outdated year of data in the Limitations or, even better, conduct the analysis with multiple years of data included

The year 2013 was the most recent year with complete data at the time the study was conceptualized. As already indicated, the data from that single year was deemed sufficient to give the requisite sample size to answer the primary question at an affordable rate without causing selection bias (as all qualifying participants would be included.) Although it can be reasonably argued that the data are ‘old’ they are not ‘outdated’ as far their relevance is concerned. The practice of using CT scan for non-trauma medical emergency remains both in our hospital as well as in large number of low- and middle-income settings as a result of lack of access to MRI that we mention in the manuscript.

MISCELLANEOUS:

-The authors should be careful not to use abbreviations unless defined with the first appearance. Also, why use the abbreviations HCP and VPS in the Conclusions?

We have noted this. The manuscript has been reviewed and amended to deal with this issue

Under Data Availability, the authors state, “Data cannot be shared publicly because of this is a paediatric study.” To my knowledge, this should not preclude them from sharing de-identified data.

This was a misunderstanding on our part. We had thought that the request was for raw data for which IRB had not given consent. All data that meet PLOS’s minimal dataset have been presented in the article, including those used in any revisions.

Reviewer 2

This is an interesting retrospective paper reviewing the indications and outcomes of CT in children with acute medical illness.

Although the data is interesting, I think the focus should be different. From my perspective as a clinician the most relevant finding is in which cases the CT led to change in management.

Thank you for the observation. We agree with your perspective and highlighting which patients received surgical intervention post CT scan was one of the aims of the study. In the Introduction we state, 

“Our study aimed to describe the clinical use of emergency head CT scans at a tertiary paediatric hospital in a low and middle-income country setting. The primary outcomes of interest were indications for and findings of head CT imaging in children presenting for acute medical care, as well as to establish baseline characteristics and interventions performed post CT scanning.” Lines 80-84

There are different indications for head CT. Analyzing these cases as a group may lead to misinterpretations and wrong management

Thank you. While we acknowledge that there are different head CT indications, the indications have been stratified to highlight these differences. We have additionally added extra columns to Tables 2 and 3 to further stratify the data such that groups with different characters are analyzed separately.

INTRODUCTION – Lines 70-71 (other effects of high-dose irradiation …) are not relevant

Thank you for the input. The sentence has been removed.

METHODS- It is important to know if the hospital has clinical guidelines in which cases a CT should be ordered

The hospital does have clinical guidelines, however these are relatively flexible. Head CT scan requests are still discussed on a patient by patient basis as they are not set in stone. We have added the following sentence to clarify this-

“Although the hospital has clinical guidelines for performing CT in the MEU, these are relatively flexible and there are frequent deviations from these guidelines.” Lines 96-97

Reviewer 3

This is an interesting study on CT scans for medical emergencies in a MLIC that has a fantastic database from which it can draw from. Overall, the data as presented is expansive, but somewhat difficult to follow. Would suggest a stronger structure framed around clinical presentations (i.e. headache, seizure, etc) as these are usually quite clear with any patient that presents with a chief complaint

Thank you for the positive feedback, it is much appreciated. Although we agree that structuring the analysis around clinical presentation would make for a rational structure, we found it more meaningful to use indications for CT scans as the starting place for our analysis. The reason for this is that it is not so much the clinical presentation that dictates whether a request for CT would be granted or not, but rather justification of why such a request is being made. That is, for each request, a justification (indication) is demanded. Often the indication combined multiple clinical features.

 It is somewhat inappropriate to lump all positive CT results regardless of clinical presentation and analyze these as a whole (i.e. the clinical indication at baseline for a head ct win a patient with a VP shunt is significantly higher than for a headache…it is like comparing apples and oranges). This was done with table 1 and 3.

We acknowledge this input. Table 1 was intended to show the baseline characteristics of the group with no analysis done. We have updated Table 3 to show the VP shunt group on its own.

Similarly, most of the data presented are primarily descriptive in nature. It seems as if there could be a bit more a deeper analysis of the rich data you have in addition to what has been down that would help a front-line physician. For example:

o Emphasis on clinical features associated with positive CT heads for each clinical presentation

o Determining the impact of CT head results (either positive or negative) on the clinical course of patients

We partly responded to this question above. The study was designed as descriptive study and not specifically designed with the requisite power for a primary hypothesis testing design. As already pointed out in our response to this reviewer, even where ‘statistical significance’ is found, the results must be interpreted with caution. We agree fully with this review perspective and believe we should in the current manuscript avoid over-analysis of the data that risks generating spurious results, both positive and negative. We have however added additional analysis as shown in Table 3 and associated text that reads, 

“ There were 29 (46.8% ) of the 62 patients with VPS that had at least one abnormal finding on CT (interval change) compared to 63 (25.3%) of the 249 without VPS (i.e. no new findings or interval change if previously scanned); P=0.001.” (Lines 190-192).

 We have also added the following sentence to the Discussion section to highlight the need of studies designed specifically for hypothesis testing, “There is a need to design properly powered studies to do more deeper analysis that look at the impact of CT head results on the clinical course of patients.” Lines 320-322

ABSTRACT:

• Re results: would be useful to be presented with more data on what proportion of children with x clinical presentation (I.e. trauma, VP shunt concerns, headache) had positive findings, and what proportion required urgent neurosurgical intervention, rather than current cumulative data a presented for all patient. It is useful to know the clinical features associated with an abnormal CT scan as presented though. Adverse event data is hard to comment as the n’s are so small and does not seem to be a primary objective of the study. Would remove from abstract

-Thank you for this fair criticism. The abstract has undergone a major revision to take into consideration all the issues raised including showing the data as suggested.

INTRODUCTION:

• Re primary objective of study (Pg4; Line80): This study is more of an observational study on CT use characteristics. It would be difficult to say it is to assess clinical utility, because it is retrospective, and not prospective. Would consider rewording.

The primary aim of the study was indeed descriptive. The sentence has been modified to say, 

 “Our study aimed to describe the clinical use of emergency head CT scans at a tertiary paediatric hospital in a low- and middle-income country setting.” Lines 80-81.

• The stated null hypothesis (Pg 5; Line83-86) is awkward given your study design. If you have a null hypothesis would expect sample size and power calculations

‘Hypothesis’ here refers to ‘assumptions’ made, rather than hypothesis testing per se. To remove confusion this sentence has been removed and the stating of the secondary outcome rephrased to now read,

 “Secondarily, we compare frequencies of presenting factors between participants with normal and abnormal findings on CT.” Lines 84-85

METHODS:

• Pg 5; Line 98: The fact that this study only assessed MEDICAL EU and excluded Trauma related emergency indications is a significant limitation to the study design and pragmatic generalizability of the results. Trauma is a primary indication and utility for CT scanning children in an Emergency setting. Please indicate why trauma patients were not included and if feasible, would consider their inclusion in a reanalysis of the data. Many institutions in an LMIC may not have separate Medical and Trauma Emergency Units and see patients as a whole. Would consider explaining this rationale in either your introduction section more clearly.

Thank you for raising this point. At this hospital there is a separate trauma unit which is not linked to the Medical Emergency Unit and indications for computed tomography (CT) in the trauma unit are according to the trauma guidelines. Trauma patients were not included in this study as the guidelines for imaging these patients are already set according to international norms. The study was designed to focus specifically on head CT scans done for medical indications as this is a grey area we felt needed to be addressed in this institution. PECARN, CATCH and CHALICE are all applicable to head trauma which was not the focus of the study as we set out to analyse the indications in a medical setting. The following section has been added to the introduction to explain the rationale, 

 “While there are clear guidelines for CT following head trauma, the same is not the case for CT head in non-trauma medical emergency.[9, 10] This means that requests for CT scan are generally made on unclear bases which has significant implications when there are significant resource limitations such as in low and middle income country (LMIC) settings.” Lines 73-76. Two additional references have been added to the section:

9. Khalifa M, Gallego B. Grading and assessment of clinical predictive tools for paediatric head injury: a new evidence-based approach. BMC Emerg Med. 2019;19(1):35. Epub 2019/06/16. doi: 10.1186/s12873-019-0249-y. PubMed PMID: 31200643; PubMed Central PMCID: PMCPMC6570950.

10. Lyttle MD, Crowe L, Oakley E, Dunning J, Babl FE. Comparing CATCH, CHALICE and PECARN clinical decision rules for paediatric head injuries. Emerg Med J. 2012;29(10):785-94. Epub 2012/02/01. doi: 10.1136/emermed-2011-200225. PubMed PMID: 22291431.

Lines 365-371 

• Re outcome measures/subgroup analysis, it would be interesting to assess practitioner adherence of practice to well established clinical decision tools for CT head (I.e. PECARN, CATCH or CHALICE).

As explained above, there are no clear decision tools currently in use outside trauma settings. We did not think it appropriate to use head trauma clinical decision tools in this context.

• Minor point: Pg6; Line113: to be specific with terminology would change to “Pearson’s Chi-square test). Would also need to see sample size and power calculations as stated above.

The amendment to the terminology has been made. The sample size calculation has been included. The following lines have now been added to the Methods section, 

“Categorical variables were compared using Pearson’s Chi-square or Fisher’s exact tests as appropriate. We estimated that over a one-year period about 250 to 350 children seen in the medical emergency unit would make inclusion criteria. Based on this sample size, for outcomes ranging from 5% to 20%, the precision will fall within 3% of the point estimate.” Lines 116-120

RESULTS:

• Table 1: would include demographic information broken down by clinical presentation (i.e. a column CT scans done for seizure, demographics of this group, etc.)

Table 1 shows baseline characteristics of the whole group. We believe that this should remain as is in order for the reader to understand the sampled population. We do not believe that stratifying on the bases of the listed variables adds any extra value. But we agree that stratification as mentioned previously is necessary to bring clarity to other parts of the manuscript and have as a consequence created an extra column in Table 3 as indicated previously.

• Table 2: would revise to include additional data including: proportion with abnormal findings; proportion of neurosurgically significant CT head findings. Would also break down seizures between generalized and focal seizures.

We have added data on abnormal findings to Table 2 but decided against adding the other things to the table as this made the table cumbersome and confusing. Data on neurosurgically significant CT head findings and seizure types are contained in text (Lines 229-232 and Lines 195-205, respectively) as well as (new) Table 5.

• Consider potentially determining sensitivity and specificity of clinical presentation with CT findings (abnormal vs normal).

While the idea of sensitivity and specificity is quite attractive to us, we fear that it communicates something that we are not sure we can sustain. A normal finding on CT will by itself not indicate that everything is normal in the presence of clinical illness. While proportions of abnormal CT’s with specific clinical signs may approximate the sensitivity of the clinical sign, absence of the same signs as well as absence of findings on CT cannot be interpreted to estimate specificity.

• Pg10; Lines196-200: Presented data is not very useful. This is mainly cumulative data and without context is hard to understand.

We felt that it would be useful to show aggregate data as an introduction to this section. We have revised this section slightly to present it in a way we hope is both succinct and more useful.

• Table 3: would be strengthened by including additional data columns on initial clinical presentation, and indications for CT head (clinical presentation). Could also include what proportion of CT findings were actually significant and resulted in a change in management

We have updated Table 3 by stratifying the data to add a column of those with VP shunts as suggested by a previous comment. We found that attempting to add more data to this table made the table cumbersome and confusing. The other data requested is contained elsewhere in the manuscript such as the new Table 5. 

• Table 4: Although statistical significance is achieved, it is difficult to know if this is valid given the likely inadequate sample size – would explicitly say this in the manuscript text somewhere (limitations?

We have included the following line to the section indicating limitations of the study, “In instances where some association has been established, the results must be interpreted with caution given the small sample size.” Lines 318-319

MANAGEMENT AND OUTCOME:

• If one patient had a severe contrast reaction associated with CT head, cannot claim no adverse reactions occurred as stated abstract.

Mention of this case has been removed in response to a previous comment that suggested that this may be a distraction to the message of the manuscript.

• This whole section is a bit oddly placed and not sure what it adds to the manuscript. Break down what proportion of patients got LPs based on clinical presentation.

The section on lumbar puncture has been largely revised and made succinct to focus it and stop it distracting from the theme of the manuscript. Lines 223 to 227 

• Is it possible for much of these data in this section to be summarized in a new table?

Table 5 has been created to accommodate some of the data.

• Overall, results section would be strengthened and made more relevant if authors restructured the manuscript around clinical presentations (I.e. Meningitis/seizure/headache) and presented the data and clinical features for each clinical presentation (under clearly marked subtitled sections). Otherwise as presented, it is merely a lot of summative data with no context and therefore no relevance and little structure that makes it difficult for readers to follow.

We have responded to the approach we took in reporting the results in a previous response. (Reviewer 3, comment 1). More analyses have been added as shown in Table 3 and related text. Lines 187-192.

DISCUSSION:

• Structure of discussion is overall much easier to follow as it is based around clinical presentations. Would still benefit from more structure as stated above.

Thank you for the affirmative feedback. 

• Pg18; line 379: I am likewise concerned with a number of statements about significance when clinical sample size is not determined. It deserves more attention rather than a tag-along sentence mentioning missing data. A major point that should also be discussed is the omission of trauma patients (a primary driver of head CT usage in an ED).

We have already responded above to the suggestion of having a clinical presentation-based structure. In general, the discussion has seen massive revisions in response to review comments that we hope gives it a better and clearer structure.

• Likewise, I would think a major limitation is your study design – it is an observational study with no overt comparator group.

Noted, we have responded to all the issues raised here elsewhere in our responses

• Pg18l; line 379: Would clearly indicate this is a limitations section.

We agree that retrospective studies are fraught with limitations which we have noted as such as we noted, “Our study is limited by its retrospective design ...” Line 315. However, we also believe that such studies provide ‘low hanging fruit’ especially in poor resourced areas to generate hypothesis and to audit practice which others can learn from at a relatively affordable rate.

MINOR POINT THROUGHOUT MANUSCRIPT:

• “Commonest” is awkward and seems a bit too colloquial [albeit a word still]. Consider “most common”. Likewise, with “commoner” and “more common”

Thank you for the feedback.

“Commonest” and commoner have been respectively replaced with “Most common” and “more common” throughout the manuscript.

---

## [Editor Report · Decision Letter 2]

14 Sep 2020

Clinical use and indications for head computed tomography in children presenting with acute medical illness in a low- and middle-income setting

PONE-D-20-01190R2

Dear Dr. Buys,

We’re pleased to inform you that your manuscript has been judged scientifically suitable for publication and will be formally accepted for publication once it meets all outstanding technical requirements.

Kind regards,

Itamar Ashkenazi

Academic Editor

PLOS ONE
---

## [Editor Report · Acceptance letter]

18 Sep 2020

PONE-D-20-01190R2 

Clinical use and indications for head computed tomography in children presenting with acute medical illness in a low- and middle-income setting 

Dear Dr. Buys:

I'm pleased to inform you that your manuscript has been deemed suitable for publication in PLOS ONE. Congratulations! Your manuscript is now with our production department. 

Kind regards, 

on behalf of

Dr. Itamar Ashkenazi 

Academic Editor

PLOS ONE